# Self-Cleaning: From Bio-Inspired Surface Modification to MEMS/Microfluidics System Integration

**DOI:** 10.3390/mi10020101

**Published:** 2019-01-30

**Authors:** Di Sun, Karl F. Böhringer

**Affiliations:** Department of Electrical & Computer Engineering, University of Washington, Seattle, WA 98105, USA; dxs535@uw.edu

**Keywords:** self-cleaning surface, superhydrophobic, superhydrophilic, superomniphobic, microfluidics, electrodynamic screen, gecko setae

## Abstract

This review focuses on self-cleaning surfaces, from passive bio-inspired surface modification including superhydrophobic, superomniphobic, and superhydrophilic surfaces, to active micro-electro-mechanical systems (MEMS) and digital microfluidic systems. We describe models and designs for nature-inspired self-cleaning schemes as well as novel engineering approaches, and we discuss examples of how MEMS/microfluidic systems integrate with functional surfaces to dislodge dust or undesired liquid residues. Meanwhile, we also examine “waterless” surface cleaning systems including electrodynamic screens and gecko seta-inspired tapes. The paper summarizes the state of the art in self-cleaning surfaces, introduces available cleaning mechanisms, describes established fabrication processes and provides practical application examples.

## 1. Introduction

A self-cleaning surface is defined as a surface that prevents or reduces surface contamination such as dust, water condensation, stains, or organic matter [1,2]. Self-cleaning surfaces have been under development at least since the late twentieth century. Related research involves multi-disciplinary backgrounds and aims at a broad range of applications including skyscraper windows, car windshields, solar panel cover glass, surveillance camera lenses, and water drag reduction on ship hulls [3]. Scientists have been inspired by nature to modify the microscopic structural and chemical properties of surfaces based on discoveries from plants, insects, and reptiles [4,5,6,7]. The approach is termed “biomimetics” as it mimics the micro/nano structures on plant leaves, insect wings, and animal skins.

Self-cleaning surfaces in nature rely often on water droplets (rain or condensation) and gravity to wash away surface contaminants. Such surfaces require to be positioned at a tilted angle, and the path that the droplet follows during cleaning is not precisely defined. Considering these drawbacks, more systematic designs have been proposed employing micro-electro-mechanical systems (MEMS) and microfluidics approaches, in combination with surface modifications for better cleaning effects. Many innovative designs have been implemented aiming at reducing labor and the overall maintenance cost for clean surfaces.

In this review paper, we discuss the working principles of different self-cleaning surfaces and systems, including both passive surface structure design and active microsystems. The design strategies and fabrication processes are introduced, as well as application examples. The paper provides guidelines for self-cleaning surface design and implementation.

## 2. Passive Self-Cleaning Surfaces

Passive self-cleaning surfaces rely on surface modifications, combining both physical and chemical changes of their surface properties. The surface energy will be altered accordingly to reduce the adhesion of a water droplet to the surface. The droplet can slide off or roll off the surface under gravity when tilted to clean the contaminants along its path. No other external physical fields are involved in dislodging the contaminants [8]. In this section, we will discuss the fundamental surface wettability theory and different surface modification approaches, including superhydrophobic surfaces, superomniphobic surfaces, superhydrophilic surfaces, and liquid infused porous surfaces.

### 2.1. Surface Wettability Theory Review

To describe the wettability properties of a surface, the static and dynamic contact angles of a sessile droplet are commonly characterized. As depicted in Figure 1a, the static contact angle (CA), θ, is determined by the tangent angle between the smooth solid surface and the liquid meniscus outline [9]. The basic law for surface wettability was first derived by Thomas Young in 1805, known as Young’s Equation [10]:
(1)$$\cos\;\theta =\frac{\gamma_{\mathrm{SG}}-\gamma_{\mathrm{SL}}}{\gamma_{\mathrm{LG}}}$$
where γ_SG_, γ_SL_, and γ_LG_ are, respectively, the interfacial surface tension at the solid/gas, solid/liquid, and liquid/gas interfaces. The model is based on the thermodynamic equilibrium approach between the three phases. Surface wettability is described as hydrophobic (CA > 90°) when the solid surface free energy in air is lower than in liquid, and hydrophilic (CA < 90°) when the solid surface free energy in air is higher than with liquid on top [9,11,12]. The suffix of “-philic” or “-phobic” describes whether the liquid has affinity or lacks affinity to the solid. A variety of contact angle measurement methods have been proposed, including direct measurement by goniometer [13], captive bubble method [14], Wilhelmy method [15], capillary tube [16], and capillary bridge [17,18,19], among others. These approaches rely on Young’s Equation and the interfacial surface tension remains unchanged during the measurement. The goniometer is the most widely used tool to measure a static contact angle. The profile of a sessile droplet silhouette is captured, and the droplet contact angle is determined by aligning the tangent of the droplet profile at the liquid/solid contact point. To analyze the droplet contact angle, we cannot cover all the methods but briefly introduce axisymmetric drop shape analysis (ADSA) [20,21,22,23,24], theoretical image fitting analysis (TIFA) [25], and high-precision droplet shape analysis (HPDSA) [26,27]. ADSA was first developed by Y. Rotenberg, et al. to minimize the squares of normal distances between the droplet sideview profile and theoretical capillary curve based on the Laplacian Equation [20]. The surface tension is an adjustable parameter and droplet profile coordinates are determined by edge detection techniques. Instead of knowing the coordinates along the droplet profile, F. K. Skinner, et al. modified the ADSA by measuring the droplet diameter from the top view [24]. The modified approach can measure low contact angles (CA < 30°). ADSA uses a one-dimensional profile curve and requires edge detection. The TIFA method determines the droplet surface tension by two-dimensional fitting between the pendant droplet image and the theoretically calculated profile without the need of edge detection. M. Schmitt and F. Heib developed the HPDSA methods to analyze droplets on inclined surfaces [26,27], using localized ellipse fitting to determine the contact angles separately for non-axisymmetric drop shapes. Sequential images of dynamic droplet contact angle change can be extracted by this method.

As the droplet dynamically wets or dewets the surface, the liquid-air-solid three phase contact line (TPL) starts to advance or recede. More than one state can exist. The interfacial energies at the TPL will have multiple energy equilibrium states [28] caused by surface imperfections such as local defects or roughness. Macroscopically, we can monitor a minimum CA value, called receding angle, θ_rec_, as the TPL recedes and a maximum CA value, called advancing angle, θ_adv_, as the TPL advances. The difference between the advancing and receding angle is called contact angle hysteresis (CAH, θ_CAH_ = θ_adv_ − θ_rec_), shown in Figure 1b,c. Due to contact angle hysteresis, a droplet can be pinned on inclined surfaces, as shown in Figure 1d. Sliding angle (SA), α, is defined as the angle between the tilted substrate and the horizontal plane when a sessile droplet starts to move down the surface due to gravity [29]. The relationship describing the sliding angle on a smooth surface with contact angle hysteresis can be described as [30]:
(2)$$\mathrm{mg}\;\sin\;\alpha /w=\gamma_{\mathrm{LG}}\;(\cos\;\theta_{\mathrm{rec}}-\cos\;\theta_{\mathrm{adv}})$$where *m* is the droplet mass, *g* is the gravity constant, and *w* is the droplet width in contact with the surface.

Large contact angle hysteresis implies strong pinning or stiction of the liquid to the surface [31]. Consequently, K.Y. Law proposed a definition of surface hydrophobicity based on the receding CA θ_rec_ instead of the static CA θ [11]. A more distinct difference between the measured wetting force and θ_rec_ could be observed when θ_rec_ > 90° or θ_rec_ < 90°. On the basis of the surface affinity measurements, the author proposed that the surface was hydrophilic when θ_rec_ < 90° and the surface was hydrophobic when θ_rec_ > 90°.

Young’s Equation does not take the influence of surface roughness into consideration. Wenzel (1936) [32] and Cassie-Baxter (1944) [33] proposed models to study the water droplet apparent CA on a rough surface. For homogeneous wetting conditions, the CA can be estimated using the Wenzel model as in (Figure 2a):
cos θ* = *r* cos θ(3)where θ* is the apparent CA on a rough surface, *r* is the surface roughness defined as the ratio of total rough surface area over the projected flat region (always ≥ 1), and θ is the Young (intrinsic) CA as defined on a flat surface. The Wenzel Equation shows that surface roughness amplifies the wetting on originally flat surfaces [34]. On hydrophilic rough surfaces, the apparent CA θ* becomes smaller than the intrinsic CA θ, while on hydrophobic rough surfaces, the apparent CA θ* becomes larger as compared to the intrinsic CA on flat surfaces.

However, on rough hydrophobic surfaces, the surface energy of a dry solid surface is lower compared to a wet liquid/solid interface. Instead wetting all solid surface asperities, the water droplet often forms composite interfaces with air pockets and solid surfaces underneath [35,36]. A model that captures this more complex heterogeneous scenario was proposed by Cassie and Baxter to predict water droplet contact angle on composite surfaces (in particular, solid and air, see Figure 2b):
cos θ* = ϕ_air_ cos θ_air_ + ϕ_solid_ cos θ_solid_(4)where ϕ_air_ and ϕ_solid_ are area fractions of the air and solid surface and ϕ_air_ + ϕ_solid_ = 1. θ_air_ and θ_solid_ are water CAs when in contact with air or a solid surface. From Young’s Equation, it follows that the contact angle of water with air is 180°, thus cos θ_air_ = −1, and we can derive the relationship between the apparent CA θ* and the Young CA θ = θ_solid_ on the composite surface as:
cos θ* = −1 + ϕ_solid_ (1 + cos θ).(5)

In this case, the solid surface region fraction ϕ_solid_ represents the portion of the heterogeneous surface in contact with liquid, as opposed to the surface roughness *r*, which is the key parameter to determine the contact angle on homogeneously wetted rough surfaces.

By studying CAs or CAHs on chemically heterogeneous surfaces, the Wenzel and Cassie-Baxter model is accurate only along the contact TPL instead of the whole contact region between droplet and surface. Experiments on chemically heterogeneous surfaces were performed by C.W. Extrand [37] and L. Gao and T. McCarthy [38]. In Gao and McCarthy’s experiments, a circular spot with different surface finish was patterned on the substrate, e.g., a hydrophilic spot on a hydrophobic field, or a flat hydrophobic spot on a rough field. By continuously expanding or retrieving the droplet, the advancing CA, receding CA and the CAH were all determined by the surface condition on the homogeneous periphery at the TPL instead of the average surface conditions beneath the droplet away from the TPL.

On a flat surface with the known lowest surface energy coatings based on the hexagonal close alignment of –CF_3_ groups, the highest contact angle of a sessile water droplet can only be approximately 120° [12]. With surface roughness, according to the Cassie-Baxter model, when ϕ_solid_ is close to zero, the apparent contact angle θ* approaches 180°. However, as shown in Figure 2c, the water can impregnate into the surface roughness structures. Studied by Miwa, et al. [39], the Cassie-Baxter Equation may be modified as:
cos θ* = −1 + ϕ_solid_ (1 + *r* cos θ)(6)where *r* is the analogous surface roughness term as in Wenzel’s Equation and *r* ϕ_solid_ represents the ratio of the substrate-water contact area to the projected surface area. Interaction energy between the liquid and solid is *r* ϕ_solid_ times higher when compared to a flat surface. A low SA (~ 1°) is achieved only with a high trapped air ratio and reduced *r*, meaning the droplet needs to rest at the tip of the roughness structures with small impregnation regions into the roughness, close to perfect Cassie-Baxter state. The water impregnation level was further studied with atomic force microscopy (AFM) on hierarchical structures together with Miwa’s model by N. Okulova, et al. [40]. Because of the water impregnation, a strong liquid–solid surface adhesion can coexist with high contact angle of the droplet on the surface, named “rose petal effect” [41]. The surface roughness in this case will increase the CA hysteresis [28]. The water droplet keeps a high CA (153°) but meanwhile exhibits a high CA hysteresis by pinning to the substrate even when the substrate is placed vertically or upside down.

Water droplets on top of surfaces with a high CA (>150°), low SA (<10°) and low CAH (<10°) are most favorable for self-cleaning. This property is termed superhydrophobicity [42,43]. On superhydrophobic surfaces, a water droplet can roll off the surface by gravity easily when the surface is slightly titled and pick up dust particles along its path. The adhesion force of dust to the superhydrophobic substrate is several times lower than on hydrophilic or even hydrophobic surfaces [44]. We term such a cleaning strategy as passive [45] and the cleaning process will happen only when the water droplet is dispensed on the tilted surfaces.

### 2.2. Superhydrophobic Surfaces

Two botanists, Barthlott and Neihuis [46], studied the microrelief of plant surfaces and discovered the papillose epidermal surface roughness and epicuticle wax coatings were the two key factors for self-cleaning mechanisms. Water droplets on top of lotus leaves kept high contact angles (~160°) and low sliding angles (< 5°), promoting the motion of the water droplets under gravity when the surface was tilted. Due to the surface roughness, dust particles on top of the leaves had reduced contact regions to the surface, which decreased the adhesion forces and were much easier to be cleaned away. A number of review articles have been published related with superhydrophobic surface fabrication processes and applications [3,47,48,49]. In this section, we have a concise discussion on the superhydrophobic surface design parameters and artificial superhydrophobic surface examples.

Inspired by the lotus leaf in nature, scientists have explored ways to mimic the lotus effect by designing micro-sized surface roughness and low surface energy coatings. Figure 2d shows the top view of a typical artificial superhydrophobic surface with square pillars. The Wenzel Equation (3) and the Cassie-Baxter Equation (4) now become [50,51]:
(7)$$\cos\;\theta_{w}*=(1+\frac{4\phi \mathrm{solid}}{\left(a/h\right)})\;\cos\;\theta$$
cos θ_c_* = −1 + ϕ_solid_ (1 + cos θ)(8)
(9)$$\phi_{\mathrm{solid}}=\frac{1}{{\left(b/a+1\right)}^{2}}.$$

From the Equations, the Wenzel state is dependent on the pillar height while the Cassie-Baxter state is not. In both states, the droplet is in a stable thermodynamic equilibrium. An energy barrier exists to prevent the transition between these two states. To be in Wenzel or Cassie-Baxter state is determined by how the droplet is formed. By calculating the energy of a drop of given volume in equilibrium on a substrate, a small a/h value (slender pillars) is suggested to obtain a robust state. A periodical (b/a) is recommended to make the droplet insensitive to energy state change. A two-tier surface roughness design with both microscale and nanoscale roughness is recommended, which provides more stable superhydrophobic state and lower contact angle hysteresis [52].

Figure 3 presents some examples. R. Furstner, et al. came up with strategies to create multiple types of superhydrophobic surfaces [53]. Shown in Figure 3a–c, silicon micro-sized pillars fabricated with X-ray lithography and followed by reactive ion etching processes, microstructured copper foil surfaces and a replica of lotus leaves using silicone molding were fabricated and characterized. All the surface designs had superhydrophobic properties. For instance, on a replica of plant surfaces, water droplets kept high contact angle (>150°) and low sliding angle (~7°). Cleaning efficiency was defined by checking the number of SEM images without contamination particles after surface cleaning with water droplets divided by the total number of SEM images taken. A cleaning efficiency of 90–95% could be achieved.

K. Koch, et al. created two-tier hierarchical structures of roughness by depositing lotus wax tubules on top of Si or lotus leaf replicas (Figure 3d) [54], achieving larger water droplet contact angle (~170°) and smaller sliding angle (1°–2°) compared with one-tier roughness structures like Si micropillars.

Figure 3e shows a nano-cone structure on a flexible Teflon substrate by oxygen plasma etching of a colloidal monolayer of polystyrene beads [55]. The wettability of the surface was controlled geometrically based on plasma treatment time as well as chemically by further gold nanoparticle deposition and silanization.

Figure 3f shows a low-cost porous structure of isostatic polypropylene (i-PP) [56]. i-PP was dissolved in the solvent mixture consisting of methyl ethyl ketone (MEK), cyclohexanone, and isopropyl alcohol, and dropped on a glass substrate. The solvent was further dried in a vacuum oven. The remaining i-PP formed a porous “bird’s nest” morphology. From atomic force measurements, the roughness of pure thin i-PP film was 10 nm RMS with a water contact angle of 104°, while the porous coating had 300 nm RMS and improved water droplet contact angle from 104° to 149°.

K. Lau, et al. [57] developed superhydrophobic surfaces by growing vertical carbon nanotube forests with a plasma-enhanced chemical vapor deposition (PECVD) process, shown in Figure 3g. To provide the stable high water droplet contact angle, the carbon nanotubes were coated with thin conformal hydrophobic poly(tetrafluoroethylene) (PTFE) by a hot filament chemical vapor deposition (HFCVD) process. Most of the superhydrophobic surfaces were made of fragile microstructures or polymeric materials, where durability could be an issue for field applications because of the harsh environment.

Y. Lu, et al. created a mechanically strong coating using an ethanolic suspension of perfluorosilane-coated titanium dioxide nanoparticles (shown in Figure 3h) [58]. Two dimensions of TiO_2_ nanoparticles (200 nm diameter and 20 nm diameter) were mixed and suspended in the ethanolic solution. The coating was able to be applied to various types of substrates like clothes, paper, or steel by spray, dip or extrusion coating processes and kept superior high water repellency after 40 cycles of sandpaper abrasion. The robustness of coating processes, substrate choice, and high mechanical strength allowed the paint to have potential applications in harsh environments.

Because of the droplet repellency and low adhesion, a condensed droplet on a chilled superhydrophobic substrate can be spontaneously removed. When the tiny droplets coalesce, the released energy can power the out-of-plane jumping of the droplet [59,60]. Such a jumping condensate process was applied for surface cleaning mechanism [61]. Inspired by cicada wings, K. Wisdom, et al. studied their wing structures and found the self-cleaning mechanism by jumping condensate process [61]. The cicada wing cuticle surface consisted of conical hydrophobic arrays, resulting in super-hydrophobicity with a water contact angle in the range of 148°–168° depending on the location. When the wing surfaces were exposed to vapor flow, the adhering particles or contaminants could be cleaned because of the water condensation process. Shown in Figure 4, the particles were detached from the surface by the water droplet’s out-of-plane jumping upon coalescence. The capillary-inertial oscillation of the merged droplet provided the required kinematic energy. The force between the jumping droplet and the particles in contact scaled with the capillary force: *f* ~ γ *R*_p_, where γ is the surface tension and *R*_p_ is the droplet radius of curvature. Due to the scaling law, for small particles, it was less favorable to remove the droplet by inertial forces like gravity, vibration, and centrifugal forces (scaled with *R*_p_^2^) or by hydrodynamic forces like wind blowing (scaled with *R*_p_^3^). The jumping condensate processes (scaled with *R*_p_) provided an advantageous mechanism to dislodge particles from the surface by overcoming adhesion forces (van der Waals force and capillary bridging force) to the substrate.

### 2.3. Omniphobic Surfaces

Water possesses a high surface tension compared with most other liquids (except for mercury). Low surface tension liquids rarely exist in nature so the naturally evolved surfaces can barely repel artificial low surface tension liquids in our daily lives [62]. According to the simple theoretical derivation, by combining the Wenzel model and Cassie-Baxter model Equations (3) and (5), we obtain the transitioning critical angle between the two states expressed as:
cos θ_c_ = (ϕ_solid_ − 1)/(*r* − ϕ_solid_)(10)where θ_c_ is the critical transition contact angle for a droplet from Wenzel state to Cassie-Baxter state [63]. By definition, we have *r* ≥ 1 ≥ ϕ_solid_, and θ_c_ is required to be at least 90° to make the transition happen because the right-hand side of Equation (10) cannot be positive [62]. For low surface tension liquids like hexane and decane, no existing natural or artificial surface coatings can achieve such a high contact angle of the liquids [64,65].

Researchers have successfully created artificial superomniphobic surfaces with the assistance of re-entrant structures [62] or doubly re-entrant structures [66,67], in which curvature is another key factor other than surface chemical composition and roughness. The key to realizing superomniphobic surfaces is that the liquid hanging between surface asperities cannot have higher contact angles than given by the intrinsic material wettability [68,69]. More specially, as shown in Figure 5a, if the advancing TPL forms a smaller contact angle, then an equilibrium state can be reached that prevents the droplet from further impalement [70]. The liquid-air interface inside the re-entrant or doubly re-entrant structure remains convex and the net capillary force generated is upward. According to Equation (4), when ϕ_solid_ is small (<6%), the surface can repel extremely wetting liquids (θ_c_* > 150° with θ ~ 0°). However, the liquid is difficult to maintain in suspension with small ϕ_solid_ because the liquid will impregnate into the rough structures without enough solid support. A doubly re-entrant structure is thus necessary with vertical, thin, and short overhangs to minimize the projected solid areas while increasing the solid fraction by vertical surfaces (side wall angle ~90°). As demonstrated in Figure 5b, on a conventional pillar-like superhydrophobic surface, a water droplet is suspended on the micropillar structure when the pillars are hydrophobic. However, for low surface tension liquid, the liquid-solid contact line overcomes this barrier and reaches the lower edge of the re-entrant structure, as shown in Figure 5c. For a completely wetting liquid, the contact line further wets down the overhang and reaches the tip of the curvature (Figure 5d). Because of the doubly re-entrant structure, the liquid-solid contact line stops wetting at the interior edge of the vertical overhangs while keeping ultra-low contact angle.

To fabricate the superomniphobic surfaces, efforts have been made to explore re-entrant and doubly re-entrant microstructure arrays. Figure 6a–c show different types of re-entrant designs. The micro hoodoo structure in Figure 6a was made by reactive ion etching of the SiO_2_ layer on top of a Si substrate followed by isotropic etching of the Si substrate using XeF_2_. The process resulted in Si pillars with SiO_2_ caps [71]. Figure 6b started with lithographic patterning on a copper substrate, followed by through-mold and over-mold electroplating to form hemispherical mound copper structures atop a photoresist layer [72]. After photoresist strip, the mushroom-like copper structure was created. Figure 6c demonstrates a nano-nail structure by using a deep reactive ion etching process to fabricate tall silicon pillars with SiO_2_ nail caps atop [73]. All the three designs required a fluoro-polymer coating as a finishing step to maintain the low surface tension required for stable fluid suspension. A vapor phase immersing deposition process was usually applied on SiO_2_ surfaces and a solution soaking process could be applied on metal surfaces. The self-assembled monolayer, terminated with the tricholorosilane group or thiol head group, formed stable covalent bond and modified the surface energy with a fluorinated tail group [74]. The silanization process was widely used for many surfaces to adjust the surface wetting behaviors [75,76,77,78].

As an alternative to lithography processes, A. Tuteja, et al. synthesized a class of fluoropolymers (polyhedral oligomeric silsesquioxane (POSS) shown in Figure 6d), with which the substrate was coated by electrospinning. The surface tension of the electrospun fiber mat could be altered by changing the mass fraction ratio of fluoro-POSS and a mildly hydrophilic polymer, thus systematically tuning the water contact angle [62,71].

Deng, et al. created a transparent superomiphobic surface using candle soot as a template, shown in Figure 6e [79,80]. The soot consisted of piles of nano carbon spheres with a diameter range of 30–40 nm. After depositing the soot on the glass substrate, a layer of silica shell was formed utilizing chemical vapor deposition (CVD) of tetraethoxysilane (TES) catalyzed by ammonia. The sample was sintered in the oven for 2 h at 600 °C to burn away the carbon cores and link the silica nano shells. The surface kept good transparency and superomiphobicity up to 400 °C.

Besides the re-entrant structures, doubly re-entrant structures have been fabricated, presenting superior surface properties as compared to re-entrant structures. Learning from smart springtail skins [66], T. Liu, et al. microfabricated structures with doubly re-entrant overhangs, shown in Figure 6f [67]. Due to its particular geometry, the surface could repel any of the existing fluids even without fluoro-polymer treatment of the final surface. Because of a pure combination of SiO_2_ and Si, the surface would also withstand high-temperature environments over 1000 °C. Derived from this process flow, metal or polymeric doubly re-entrant omniphobic surfaces were successfully fabricated as well.

### 2.4. Superhydrophilic Surfaces

Superhydrophobicity is not the exclusive strategy to realize self-cleaning functionality, which can also be realized while the water droplet contact angle atop a surface is extremely low (close to zero). The simplest way to increase the surface hydrophilicity is by oxygen plasma treatment, as demonstrated by B. Gupta, et al. [81]. Their process only modified the surface properties without altering the bulk substrate material. Experiments proved that the treated surface had anti-fogging and anti-fouling properties, but the hydrophilicity would decrease over time [82].

Another approach was to take advantage of both the light-induced superhydrophilicity [83,84,85] and the photocatalytic properties of TiO_2_ thin films, namely the “Photo-Kolbe” reaction [82,86]. The as-prepared TiO_2_ surface water contact angle is ~72°. The UV exposure creates oxygen vacancies at bridging sites favorable for dissociative water adsorption (Ti^3+^ sites instead of Ti^4+^ sites), making the water contact angle close to 0°. Microscopically, after UV radiation, the TiO_2_ surface wettability is not heterogeneous anymore, and the hydrophilic regions are distributed across the surface with area sizes in the sub-micrometer range, based on measurements by friction force microscopy. Macroscopically, the water will spread on the surface instead of forming droplets, to wash away surface contaminants easily [83].

The photo induced oxidation/decarboxylation/fragmentation of organic acids is well-known for photo-semiconductors like TiO_2_ or ZnO [87,88,89,90,91,92]. The TiO_2_ preparation can use wet chemical processes like sol-gel, dip-coating, or spin-coating processes [93,94]. A post calcination process is usually required to improve the adhesion between the TiO_2_ film and the substrate [95]. Upon UV radiation (< 385 nm) of the TiO_2_, the proton with an energy exceeding the bandgap would excite an electron (e^−^) from the valence band to the conduction band, leaving a hole (h^+^) on the valence band. Valence band holes react with the water through a strong oxidization process on the surface to produce reactive hydroxyl radicals (·OH) and convert surface contaminants, especially organic residues, into byproducts like water or CO_2_ [94]. Because of the weakening of the bonding, the surface contaminants are easily washed away by rain.

### 2.5. Slippery Liquid-Infused Porous Surface (SLIPS) Surfaces 

Solid substrates have been modified to create superhydrophobic or superhydrophilic surfaces by etching of physically rough texture or by chemical modification. However, Wong, et al. developed a system to create a liquid repellant surface, naming it “slippery liquid-infused porous surface” (SLIPS) [96]. Inspired by the Nepenthes pitcher plant [97], Figure 7a shows the fabrication process of the SLIPS surface. A porous solid surface was infused with low surface tension and chemically inert lubricating liquid, which wicked into the porous substrate while being immiscible and repelling to the test liquids applied to the surface. The contact angle hysteresis for sessile water drops was as low as 2.5° and the sliding angle was smaller than 5°. Figure 7b,c demonstrate the outstanding anti-fouling performance of the SLIPS surface by applying crude oil and human blood. In comparison with superhydrophobic surfaces and hydrophilic surfaces, no stains were left on the surface. Both oil and blood would quickly slip away from the SLIPS surface. Besides the superb repellency, the SLIPS surfaces also have self-healing properties [96]. Because of the surface ultra-smoothness and lack of nucleation sites [98,99,100,101], no frost formation or a reduced ice adhesion were observed on cold SLIPS surfaces.

With regards to bio-fouling applications, extensive studies have been performed on superhydrophobic surfaces [102,103,104,105,106,107]. However, the anti-biofouling property of superhydrophobic surfaces could be short-lived as the air-bubble layer trapped between the liquid and the rough surface is not stable and may disappear within several hours [108]. More bacterial adhesion could end up on the superhydrophobic surfaces due to the high surface roughness when compared with intact smooth surfaces. Extensive work has been explored by adopting SLIPS surfaces to prevent bio-fouling issues by various fabrication methods, which were more promising and with better performance than superhydrophobic surfaces. A. Epstein, et al. adopted SLIPS surfaces to prevent surface bio-film attachment [109]. Shown in Figure 8a, the SLIPS surfaces were fabricated with porous fluoropolymer substrates (with pore size of 0.2 μm). By staining the surface with bacterial culture solution, the SLIPS surface can reduce the cell attachment compared with superhydrophobic surfaces. The coffee ring effect of the biofilm was suppressed on SLIPS by leaving only a pellet of bio-stains after evaporation. Similar liquid infused porous substrate structures were obtained by phosphoric acid etching of enamels [110].

D. Leslie, et al. created a SLIPS surface with self-assembled monolayers (SAM), shown in Figure 8. The structure was applied on a wide range of smooth medical device surfaces, which repelled flowing blood and prevented thrombosis [111]. A molecular tethered perfluorocarbon (TP) layer was first coated on the smooth surfaces by soaking the plasma treated surface in liquid solution. Then a mobile layer of perfluorodecalin (LP) was applied, forming a tethered-liquid perfluorocarbon (TLP) surface. By exposing the uncoated and TLP coated acrylic surfaces to fresh human blood, the TLP surface had 27-fold less platelet adhesion and platelets were considered as one of the major components causing thrombosis. Both in vitro and in vivo experiments showed promising results, demonstrating that the TLP surfaces were resistant to the physiological shear stress brought by the blood flow while reducing the protein adhesion and thrombosis for at least 8 h.

Beyond the silane liquid soaking process to create the TP layer, M. Badv, et al. improved the hydrophobic salinization process with a more robust, reproducible and less disruptive chemical vapor deposition (CVD) process in vacuum [112,113] (Figure 8c). Coronary catheters were treated by both two silanization processes, followed by adding perfluorodecalin or perfluoroperhydrophenanthrene to make TLP surfaces. The CVD treated surfaces provided better anti-thrombotic performance compared with silane liquid solution soaking processes. As shown in Figure 8c, CVD treated catheters surfaces found no blood clot or protein adhesion after blood immersion. By mixing different self-assembled monolayer silanes (aminosilane and fluorosilane) during the surface treatment, tunable cell repellency and selective binding of antibodies can be realized. The target anti-bodies would be anchored by the aminosilanes while the fluorosilane will repel the non-desired cells, proteins or plasma clotting assays, creating the bio-functional lubricant-infused surfaces (BLPS) [114].

SLIPS can be fabricated on porous micropillar arrays with sharp overhang structures [115]. As shown in Figure 8d, the liquid on top of such surfaces meets a new liquid-air interface, compared with solid-air interfaces of the normal superhydrophobic or superomniphobic designs without liquid infusion as discussed above. The micropillar arrays with sharp overhang structures and nano-porous micropillar top surface finish were created by direct laser writing, which can process any arbitrary 3D components with sub-micrometer resolution. A layer of Al_2_O_3_ by atomic layer deposition was coated on the outer layer of polymeric micropillars and fluorinated by SAMs. Low surface tension fluid was dropped directly on the micro-pillar porous surfaces and confined by the micropillar surface roughness as well as the overhangs. The composite surface designs can repel low surface tension fluids while reducing more than twice of the adhesion force, as measured with scanning droplet adhesion microscopy.

## 3. Active Self-Cleaning Microsystems

Besides employing passive surface modification techniques, microsystems can be designed to actively remove unwanted surface contaminants or fluids [8]. Many strategies have been tested using surface tension gradients, electrostatic fields, and vibrations. Moreover, geckos can clean their feet dynamically while naturally walking with hyperextension. In such systems, water droplet movement or dust removal can be accomplished in a systematic way while applying more controlled forces. Thus, the active self-cleaning approach can be utilized in combination with passive surface modification to improve cleaning efficiency. In this part, we will first introduce the surface cleaning strategies by combining the superhydrophobic and SLIPS surface design with droplet manipulation. Then we will discussion surface dust removal techniques by electro-dynamic screen, repelling surface contaminants by high alternating voltage. At last, self-cleaning synthetic adhesives inspired by gecko setae structures are discussed.

### 3.1. Self-Cleaning Surfaces by Water Droplet Transport

Microfluidic systems have been developed using MEMS technology and widely applied for biomedical and chemical applications. The recent development of microfluidic systems using micro- or nano-liter water droplet transport, commonly known as digital microfluidics (DMF), offers the potential for a wide range of applications [116]. To control the water droplet transport, researchers have focused on creating surface tension anisotropy at the interface of gas, liquid and solid, defined as the three-phase contact line (TPL). DMF systems can be used to direct water droplet transport along the surface using chemical gradients [117], thermal gradients [118], electrowetting-on-dielectric (EWOD) [119,120,121], surface acoustic waves [122], and micro textures [123,124]. Dust particles or undesired fluids along the path of the water droplet movement can be carried away to other locations, leaving the desired surface regions clean and functional.

A typical EWOD setup is shown in Figure 9a. A water droplet is initially placed on a hydrophobic insulator surface. When a voltage is applied between the droplet and the electrode underneath, the electrostatic field will significantly modify the solid-liquid interfacial tension, leading to a reduction of contact angle and an improved wetting of the droplet on the solid surface. This effect of the voltage can be quantified by the following Equation:
γ_SL,*V*_ = γ_SL_ − ½*CV*^2^(11)where the original solid-liquid surface energy γ_SL_ is modulated by the electrostatic field (given by the normalized capacitance *C*, measured in C/m^2^, and the applied voltage *V*) to produce the effective surface energy γ_SL,V_. This leads to a generalized form of Young’s Equation (1):
γ_SG_ = γ_SL_ − ½*CV*^2^ + γ_LG_ cosθ*_V_*(12)and the Young–Lippmann Equation:
γ_LG_ cosθ*_V_* − γ_LG_ cosθ = ½*CV*^2^(13)where θ*_V_* is the effective contact angle under an applied voltage *V*.

Asymmetric interfacial surface tension change at the droplet-substrate interface can be introduced by energizing different electrodes, and the surface on top of the energized electrodes tend to be more hydrophilic. The droplet can be transported precisely controlled by sequentially enabling different electrodes. Depending on the application, two popular EWOD configurations are often used, shown in Figure 9b, the parallel-plate system, and Figure 9c, the co-planar system. For the parallel-plate system, the water droplet was sandwiched in between the top and bottom electrodes, insulated by dielectric layer (SU8, SiO_2_ [120] or parylene [125]) and hydrophobic (Teflon^TM^ AF [126,127] or Cytop^®^ [128,129] coatings. The electrode on one plate was patterned and the electrode on the other was fully grounded. Once the electrode was energized, the droplet was first deformed by the electrostatic field and driven by pressure gradient inside the droplet [130]. The parallel-plate system can prevent droplet evaporation and is less sensitive to gravity influence, compared with the co-planar system, where the cover plate is removed. However, the co-planar system has broad applications and can be integrated into many other systems which do not permit a top cover plate [131]. Meanwhile, the dielectrics and top coatings of the EWOD system can be easily integrated with superhydrophobic surface or SLIPS surface designs to enable the active cleaning capabilities by systematically controlling the droplet.

Latip, et al. explored the anti-fouling properties by applying EWOD top coatings with hydrophobic (Cytop) or superhydrophobic (NeverWet^®^) materials [132]. Different concentrations of protein solutions were prepared. A similar test bench setup as shown in Figure 9a was performed to test the contact angle hysteresis by gradually increasing and then reducing the voltage within a period of time. Compared with superhydrophobic surfaces, the contact angle hysteresis greatly increased on Cytop surfaces with increased protein concentration, maximum applied voltage and the period of time with voltage applied. Higher roll-off angle and afterwards higher fluorescence intensity with labelled protein were observed on the Cytop surface, showing a stronger protein adhesion to the Cytop. As for the droplet transport, both closed parallel plate configuration (Figure 9b) and open coplanar configuration were tested with superhydrophobic coatings. On the coplanar system, a droplet of 35 μL was applied, and the actuation was difficult to control since the droplet continued to roll on the surface due to low friction. However, in the parallel plate system, a droplet with only 5 μL was needed and was successfully transported, merged or mixed.

M. Jönsson-Niedziółka et al. showed droplet transport with a parallel plate system configuration to remove bio-particles [133]. The top and counter electrodes were separated by 300 μm spacers. A square wave voltage was applied to the selected base electrodes at the frequency of 1 kHz and the switching time between adjacent electrodes was adjusted based on droplet movement speed. The water droplet displacement was driven by the surface wettability change induced by the voltage. The cleaning efficiency was defined as: %efficiency = (1 − *N*_in_/*N*_out_) × 100, where *N*_in_ is the average number of particles inside the water droplet pathway and *N*_out_ is the average number of particles outside the water droplet pathway. Examples of water droplet transport along the electrodes and cleaning of the surface are presented in Figure 10. Synthetic particles like polystyrene latex microspheres and bio-particles, including proteins, bacterial spores, and viral simulant were tested with the system. When the substrate surface was designed to be superhydrophobic, more than 90% of cleaning efficiency could be reached with water droplets even for protein particles, which usually have high adhesion to the substrate and are hard to clean.

Y. Zhao, et al. developed a similar EWOD system for sampling of micro particles (Figure 11) [134]. The actuation electrodes were insulated with a dielectric layer (SiO_2_) and coated with hydrophobic (Teflon) coatings. Driven by sequentially actuated electrodes, the water droplet swept along the surface and picked up particles. The path covered by the water droplet became clear to visual inspection, meaning that most of the particles were collected by the moving water droplet.

An EWOD system can also be designed to remove unwanted small amounts of water residue adhering to surfaces. K. Y. Lee et al. developed an open coplanar EWOD system without a top cover plate targeting miniature camera surfaces for automobiles [135]. The electrodes were fabricated with indium tin oxide (ITO), which is transparent and can be integrated with the camera lens as a lens cover. 1–70 µL water droplet sizes were tested with different threshold voltages under surface inclination angles from 0° to 180°. Figure 12 shows a demonstration of water droplet removal as well as micro-particle removal on the camera lens cover.

Besides superhydrophobic surfaces, H. Geng and S.K. Cho combined the SLIPS with an open coplanar EWOD system [136]. The dielectric layer was SU8 and the top coating was replaced with porous fluoropolymer film infused with lubricating fluid in this configuration. Droplets can be transported along the SLIPS under voltage actuation. Bovine serum albumin (BSA) protein solution left tiny stains after evaporation on SLIPS while a large “coffee ring” bio-stain pattern was left on hydrophobic coatings. The bio-stain could be cleaned by droplet actuation as shown in Figure 13.

As an alternative to the EWOD approach, self-cleaning surface systems using water droplet transport have been realized by anisotropic ratchet conveyors (ARC) under orthogonal vibration [123,137]. Micro-scale hydrophilic semi-circular rungs are patterned on a hydrophobic background, as shown in Figure 14 above [124]. The portion of the water droplet edge that aligns with the rung curvature, which has a mostly continuous TPL, is denoted as the leading edge of the droplet, while the other portion, which has only intermittent TPLs across different rungs, is called the trailing edge of the droplet. During each vibration cycle, the leading edge provides higher pinning force than the trailing edge as the footprint of the water droplet expands and recesses. This asymmetry in pinning forces causes water droplets to move toward the direction of the rung curvature. For a surface cleaning demonstration based on the ARC approach [138], two ARC tracks were laid out in a zig-zag pattern. The white contaminant on the surface consists of powdered sweetener. 10 μL water droplets are applied to the surface and remove all the powder along their paths. Most water-soluble materials (like salt and sweetener) plus low surface adhesion insoluble particles (like sand) can be effectively cleaned from the self-cleaning ARC surface with water droplets.

### 3.2. Self-Cleaning Surfaces by Electro-Static Charge

Electrodynamic screen devices have been developed to remove dust particles for scenarios where the water resource is scarce or not available, as in desert regions. The concept of transporting particles using an electrostatic traveling wave was first developed by Masuda [139], where a series of electrodes were connected to the AC source to serve as contactless conveyors. Mazumder, et al. developed an electrodynamic screen (EDS) with traveling-wave AC field to create a self-cleaning system for the problem of dust accumulation both on Mars missions [140,141,142] and on terrestrial solar panels [143]. Figure 15 demonstrates a typical EDS design with dust accumulation and cleaning effect before and after the AC voltage was supplied [144,145]. Interdigitated electrodes were fabricated on printed circuit board or glass substrates. The electrodes were insulated with a layer of transparent polymer. As AC voltage was applied (700~1000 V peak-to-peak), the electrodynamic force applied to the particles overcame gravity and the viscous force of air to lift the particles from the surface and transported them to different locations. Over 90% of cleaning efficiency could be achieved by optimizing the frequency, voltage, and signal shape. The power consumption and the cleaning time were only in the order of milliwatts and tens of seconds. Other EDS systems were designed employing a standing-wave AC field, with simplified electrical circuit designs and high voltage resources. Bing Guo, et al. systematically studied the EDS efficiency in terms of EDS dimension size, dielectric cover thickness, dust loading level, dust deposition methods, and particle size distribution for solar energy applications [146,147]. A dust removal efficiency of 90% could be achieved within 10 s of energizing at dust loading level of 100 g∙m^−^^2^ with a voltage level of 6 kV_pp_. Dust removal efficiency improved with increased dust loading levels, reduced dielectric cover thickness and large dust agglomerations.

### 3.3. Self-Cleaning Surfaces by Gecko Tape

Geckos have attracted the attention of researchers for many years due to their ability to climb up smooth vertical surfaces. The gecko’s foot has millions of hairs, named keratinous setae, providing large van der Waals adhesive forces [148,149] that prevent the gecko from falling off from smooth vertical surfaces. Geckos have intimate contact on various surfaces with their sticky toes but their setae virtually always keep clean and dry [150]. W. R. Hansen and K. Autumn [151] studied the gecko’s keratinous setae and found the self-cleaning mechanism: Each of the millions of setae on the gecko’s toe pads has hundreds of spatulae, sub-micron triangular structures aligned in parallel with each other but not normal to the toes. An imbalance exists between the adhesive force of one or more spatulae to the dirt particles and the dirt particles to the substrate surface. When touching the clean substrate surface, the dirt has higher contact areas to the surface and tends to stick to the surface rather than the gecko’s spatulae. The adhesive and shear force of a contaminated gecko’s foot is recovered gradually by successive steps on a clean surface. By comparing the SEM images of spatula arrays after dirtying with microspheres and after several simulated cleaning steps, most of the spatula surfaces were free of micro sphere contamination. The shear force measurement also showed the gradual restoration as the simulated step numbers increased.

Self-cleaning adhesive tapes have been developed using carbon nanotubes [152] and polymer microfibrillars [153] mimicking gecko setae. Figure 16a shows contaminated polypropylene fibrillars fabricated by a thermal casting process. An estimated 42 million fibrillars were created per square centimeter with an average length of 18 µm and average radius of 18 nm. After 30 contacts on a clean glass substrate with standard simulated gecko steps (Figure 17), 60% of the Au microspheres were removed from the tip of the micropillars (Figure 16b). The sheer force could be restored by 33% after 20–25 cleaning steps. As a comparison, a conventional pressure sensitive adhesive (PSA) went through the simulated steps. The PSA surface was almost completely covered by the Au microspheres, shown in Figure 16d.

## 4. Conclusions

Self-cleaning surfaces can have a broad range of applications from bio-fouling in medical instrumentation to building and vehicle windows to solar panel cover glass in the outdoor environment. The examples mentioned in this review article provide suggestions and protocols for designing and characterizing self-cleaning surfaces and systems. Compared with passive superhydrophobic or superhydrophilic surface designs, active cleaning systems can perform the cleaning with more delicate control of water movement, and more efficient use of water resources. Both dust and unwanted water residue can be removed at the same time. However, more complex mechanical components or control circuitry are often required for active self-cleaning systems, leading to higher initial hardware investment, larger maintenance costs, and longer payback time.

Many commercial products have emerged in the market including superhydrophobic coating sprays, photocatalyst coated window glass, and solar electrodynamic shields using EDS designs, among others. The obstacles and challenges for self-cleaning surfaces currently involve poor durability and high cost in terms of scaling and mass production. The fine micro or nano structures often cannot withstand the harsh outdoor environment for an extended period of time while the polymeric coating or surface infusion fluid will age and decay over time under solar radiation. The practical lifetime for a self-cleaning surface or system might only last from months to 1 or 2 years but the requirement is usually in the 10–20 year range, especially for applications in outdoor environments, for example in dry desert regions to reduce dust accumulation, or under water to prevent bio-fouling on a ship hull. We need to strive for the creation of self-cleaning coatings for surfaces or systems with multiple merits including low cost, good scalability, durability, transparency, and antireflection.

## Figures and Tables

**Figure 1 micromachines-10-00101-f001:**
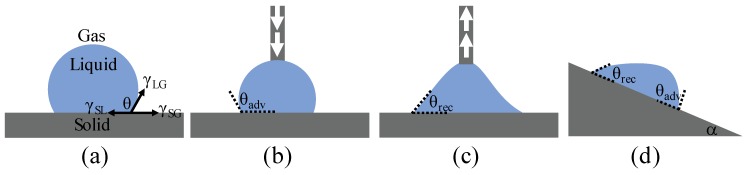
Schematics of contact angle types. The grey region represents the solid surface and the blue color represents the liquid on top. (**a**) Static contact angle θ and interfacial surface tension γ according to Young’s Equation. (**b**,**c**) represent a method to measure the advancing and receding contact angle. The arrow represents the direction of external pressure to dispense water onto or retreat water from the solid surface through a dispensing needle. (**d**) Inclination angle α, advancing angle θ_adv_, and receding angle θ_rec_.

**Figure 2 micromachines-10-00101-f002:**
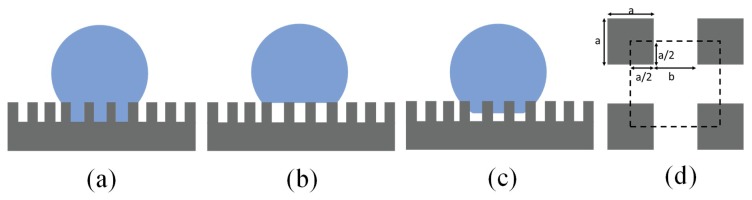
Schematics of different wetting states. (**a**) Wenzel state. (**b**) Cassie-Baxter state. (**c**) Transitional state between the Wenzel and Cassie-Baxter state, including the “petal effect” with simultaneously high contact angles (CA) and high Sliding angle (SA). (**d**) Top view of a typical artificial superhydrophobic surface design by creating surface roughness with pillars. The pillar height is *h*, the pillar breadth and width are *a* and the distance between adjacent pillars edges is *b*. The dotted square shows a periodic structure for calculation with a quarter of pillar surface counted at each corner.

**Figure 3 micromachines-10-00101-f003:**
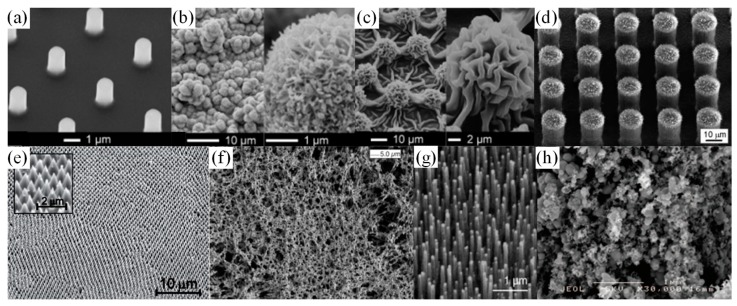
Artificial superhydrophobic surface examples imaged by scanning electron microscopy (SEM). (**a**) Micro-spikes on Si substrates. Reproduced with permission from [53], published by ACS Publictions, 2005. (**b**) Heavily structured copper film surface [53]. (**c**) Silicone rubber replicates of Alocasia structure through molding [53]. (**d**) Hierarchical structures using Si micropillars covered with lotus wax tubules. Reproduced with permission from [54], published by Royal Society of Chemistry, 2009. (**e**) Teflon nano cone arrays. Reproduced with permission from [55], published by ACS Publications, 2014. (**f**) Porous isostatic polypropylene (i-PP) structures from solution drying. Reproduced with permission from [56], published by Science, 2003. (**g**) Carbon nanotube forest grown by plasma-enhanced chemical vapor deposition (PECVD). Reproduced with permission from [57], published by ACS Publications, 2003. (**h**) TiO_2_ particles paint. Reproduced with permission from [58], published by Science, 2015.

**Figure 4 micromachines-10-00101-f004:**
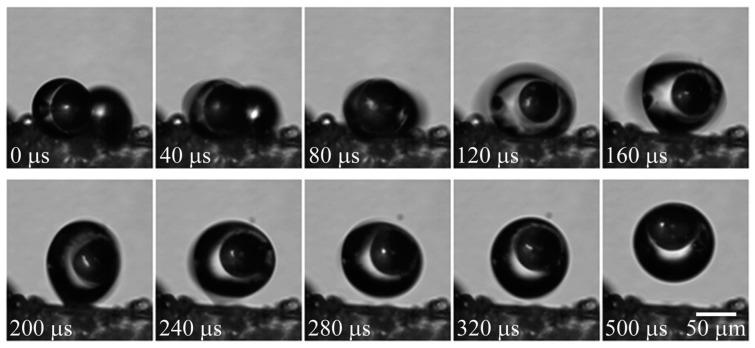
Water vapor condenses and spontaneously jumps off a cicada wing surface, encapsulating 50 µm glass beads. Reproduced with permission from [61], published by PNAS, 2013.

**Figure 5 micromachines-10-00101-f005:**
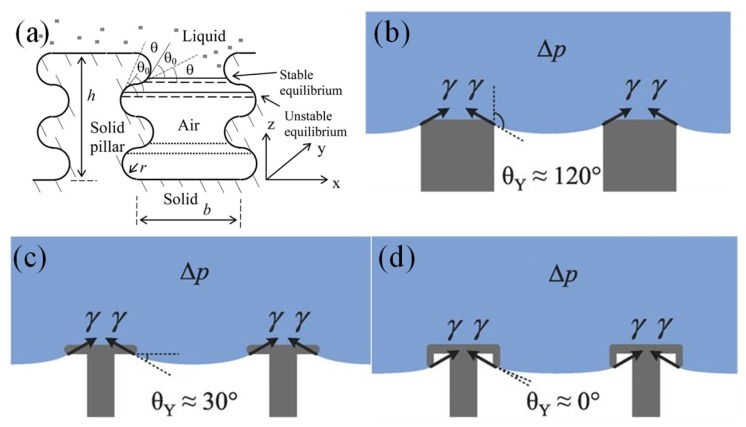
Liquid-solid contact angle required for stable liquid suspension on (**a**) Surfaces with both semicircular bumps and grooves. Reproduced with permission from [70], published by ACS Publications, 2007. (**b**) micro-pillar structures, (**c**) re-entrant structures, (**d**) doubly re-entrant structures. Reproduced with permission from [67], published by Science, 2014.

**Figure 6 micromachines-10-00101-f006:**
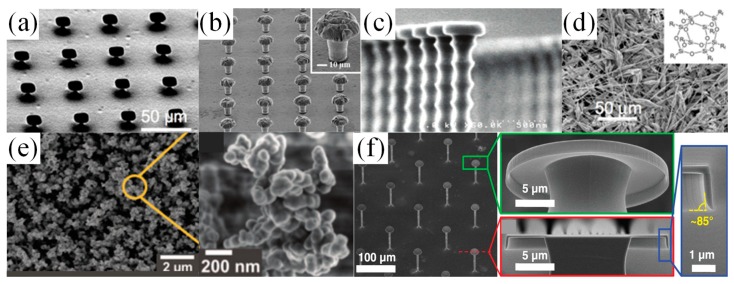
Examples of superomniphobic surface designs. (**a**) Micro hoodoo structures with a rectangular SiO_2_ cap on Si micro-pillars. Reproduced with permission from [71], published by PNAS, 2008. (**b**) Mushroom structure of copper surfaces. Reproduced with permission from [72], published by Nature, 2015. (**c**) Nano-nail structures. Reproduced with permission from [73], published by ACS Publications, 2008. (**d**) Fluorinated electrospun fibers [71]. (**e**) Candle soot structure after being coated with silica nanoshell and after carbon core removal by high-temperature sintering. Reproduced with permission from [79], published by Science, 2012. (**f**) Microposts with doubly re-entrant overhangs. Reproduced with permission from [67], published by Science, 2014.

**Figure 7 micromachines-10-00101-f007:**
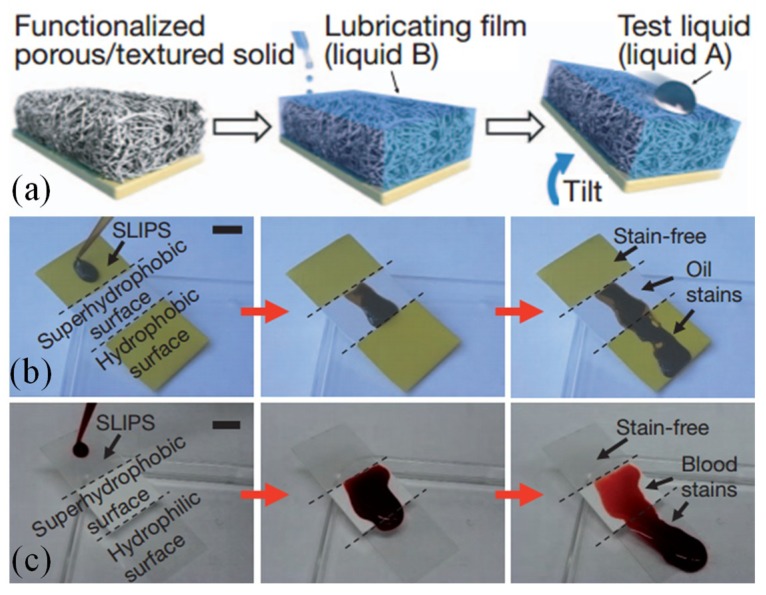
(**a**) Slippery liquid-infused porous surface (SLIPS) fabrication process flow. Low surface energy, chemically inert fluid was infused into the porous solid substrate. The surface remained smooth with lubricating film between the substrate and the applied liquid. (**b**) Crude oil and (**c**) blood movement on SLIPS, superhydrophobic and superhydrophilic surfaces. Reproduced with permission from [96], published by Nature, 2011.

**Figure 8 micromachines-10-00101-f008:**
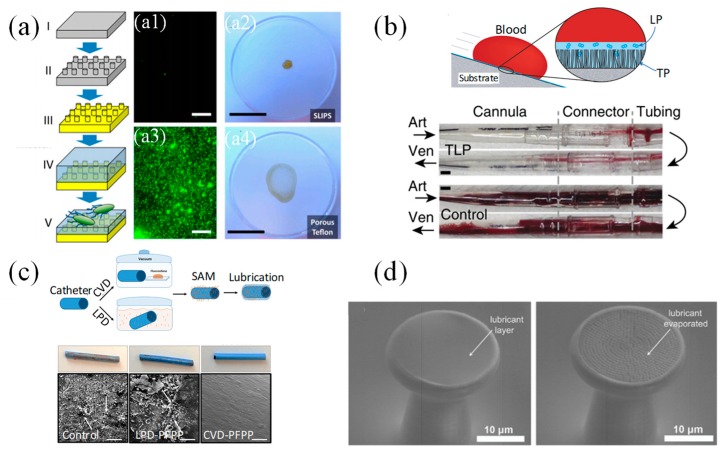
(**a**) Fabrication process of SLIPS surface using porous substrate and bio-fouling experiment with and without liquid infusion. Reproduced with permission from [109], published by PNAS, 2012. (**b**) SLIPS surface fabricated with liquid soaking deposited self-assembled monolayers (SAM). Reproduced with permission from [111], published by Nature, 2014. The experimental results showed TLP modified tubing and control tubing after 8h of blood flow. The blood flow through arterial (Art) or venous (Ven) cannula was indicated by the black arrow. (**c**) SLIPS surface fabricated with both liquid soaking and vapor deposited SAM. Reproduced with permission from [112], published by Nature, 2017. (**d**) SEM images of SLIPS and doubly re-entrant superomniphobic composite structures. The left image shows the surface with lubrication and the right image shows the surface after lubricant evaporation. Reproduced with permission from [115], published by Wiley Online Library, 2018.

**Figure 9 micromachines-10-00101-f009:**
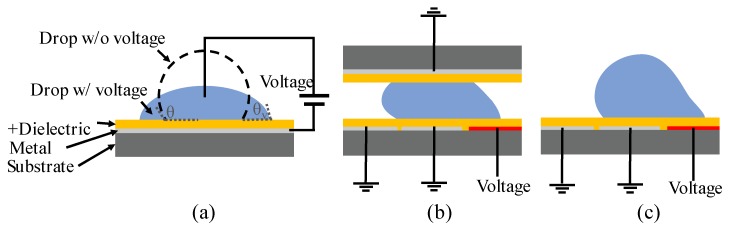
Electrowetting-on-dielectric (EWOD) setup: (**a**) Typical EWOD setup to measure contact angle change by external voltage. (**b**) Parallel-plate EWOD design to transport droplet. (**c**) Open coplanar EWOD design. The energized electrodes are marked as red color in (**b**,**c**).

**Figure 10 micromachines-10-00101-f010:**
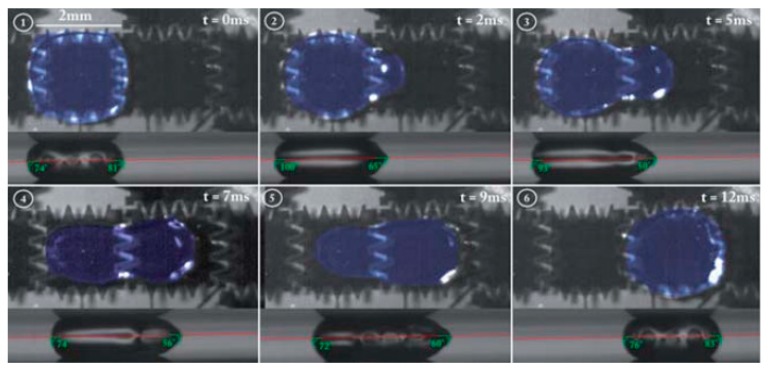
Both top and side views of water droplet transport on the superhydrophobic surface at different times. Reproduced with permission from [133], published by Royal Society of Chemistry, 2011.

**Figure 11 micromachines-10-00101-f011:**
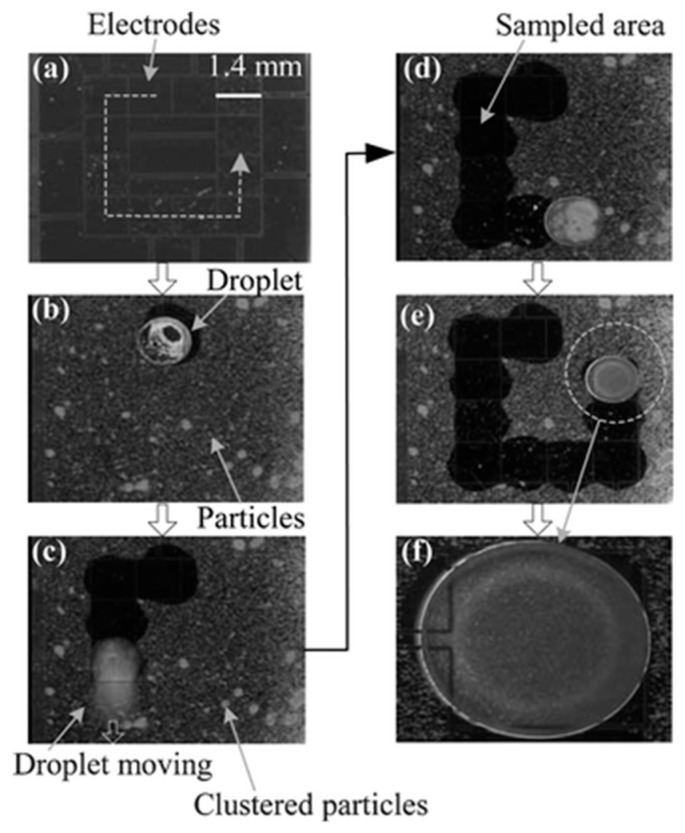
(**a**–**e**) Glass beads are collected sequentially using water droplets. The glass beads are suspended inside the sweeping water droplets. The dashed line in (**a**) indicates the path of the droplet for particle sampling and cleaning. Reproduced with permission from [134], published by Royal Society of Chemistry, 2006. (**f**) shows a close-up view of the droplet with suspended sampled particles.

**Figure 12 micromachines-10-00101-f012:**
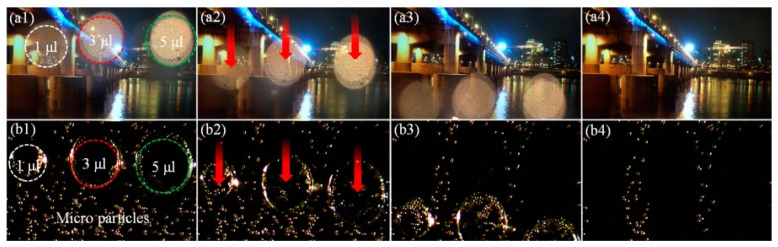
Sequential images of cleaning of (**a**) water droplets and (**b**) particles with different volumes on the lens cover of a smartphone camera. Reproduced with permission from [135], published by Elsevier, 2017.

**Figure 13 micromachines-10-00101-f013:**
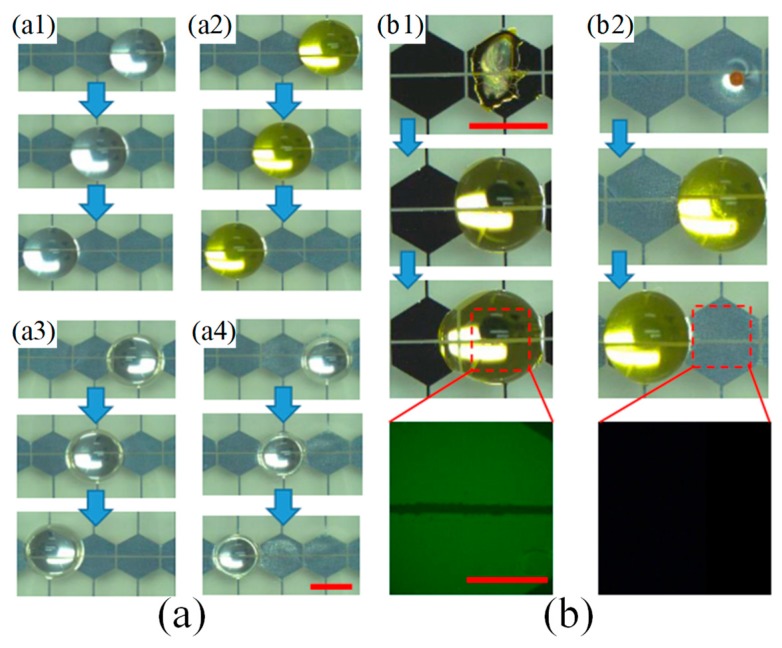
(**a**) Different liquids transported on EWOD-SLIPS surfaces: (a1) Deionized (DI) water, (a2) Bovine serum albumin (BSA) solution, (a3) propylene carbonate, (a4) isopropyl alcohol. (**b**) A droplet cleaning bio-stains left by evaporation: (b1) droplet fails to move on hydrophobic coatings due to bio-fouling, (b2) droplet moves and cleans the bio-stain. The green in the fluorescent images indicate the BSA residues on the surfaces. Reproduced with permission from [136], published by IEEE, 2018.

**Figure 14 micromachines-10-00101-f014:**
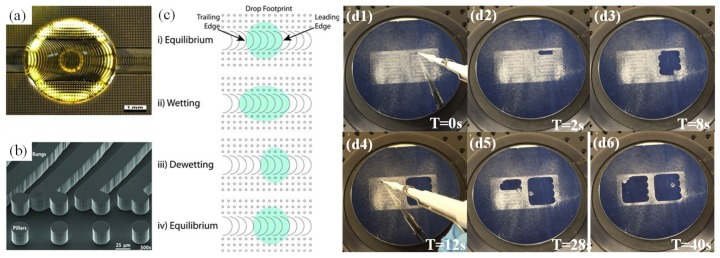
(**a**) A water droplet resting on the anisotropic ratchet conveyors (ARC) surface with etched pillars. (**b**) SEM image of the ARC surface. (**c**) Schematic of droplet interaction with droplet under vibration. The leading edge conforms to the semi-circular rung, which acts as a wetting barrier, while the trailing edge has only intermittent contacts with the rung pattern. Reproduced with permission from [124], published by ACS Publications, 2012. Reproduced with permission from [137], published by Wiley Online Library, 2012. (**d**) Surface cleaning performance for powdered sweetener (dextrose, maltodextrin, and sucralose) contamination on a chemically flat ARC surface consisting of two loop tracks. Reproduced with permission from [138], published by IEEE, 2017.

**Figure 15 micromachines-10-00101-f015:**
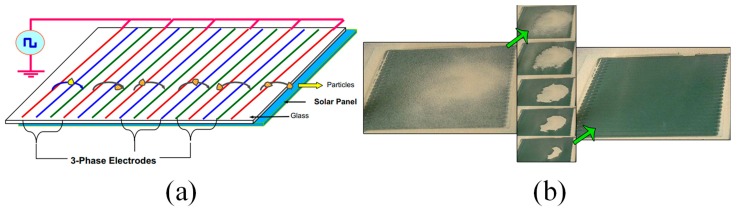
(**a**) Schematic view of a typical electrodynamic screen (EDS) design. Reproduced with permission from [144], published by Elsevier, 2013. (**b**) Sequential images of the dust removal processes on top of the EDS panel by electrodynamic force. Reproduced with permission from [143], published by IEEE, 2013.

**Figure 16 micromachines-10-00101-f016:**
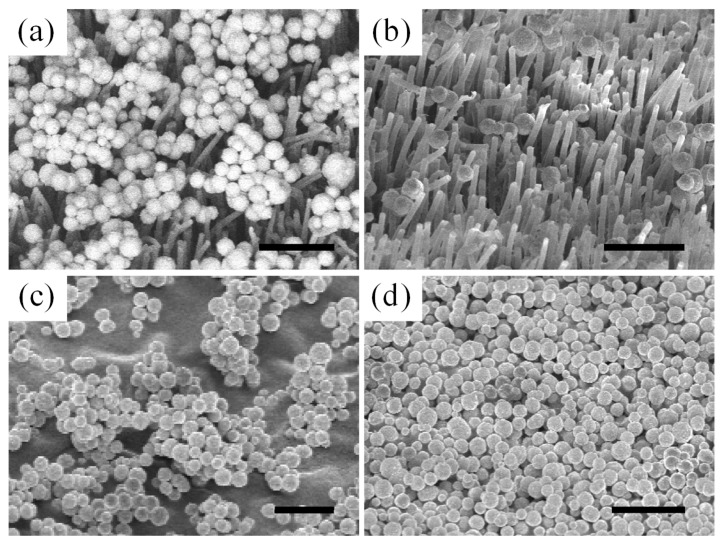
(**a**) Micro fiber adhesive contaminated with gold microspheres. (**b**) Micro fiber adhesive after 30 contacts (simulated steps) on a clean glass substrate. Some of the microspheres were trapped inside the micro fibers. (**c**) Conventional pressure sensitive adhesive (PSA) surface contaminated with microspheres. (**d**) PSA surface fully covered by the Au microparticles after the same simulated steps. Reproduced with permission from [153], published by ACS Publications, 2008.

**Figure 17 micromachines-10-00101-f017:**
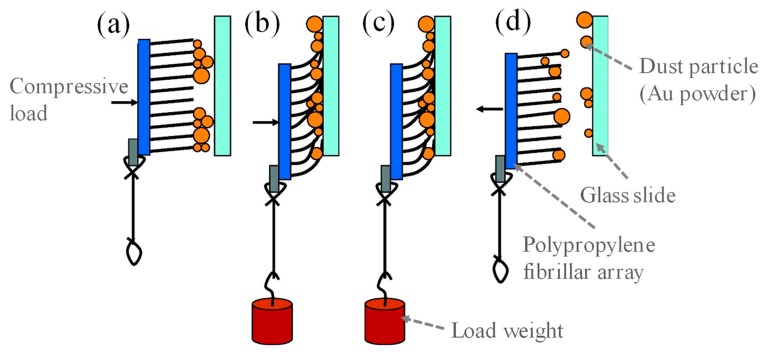
Standard protocol of mimicked gecko step cycle: (**a**) Normal compressive force was applied on the back side of the fiber substrate. (**b**) Applying shear load added to the compressive force. (**c**) Removing the compressive force to make the load a pure shear force. (**d**) Detaching the sample from the clean surface. Reproduced with permission from [153], published by ACS Publications, 2008.

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
