# Peer review of "Self-Cleaning: From Bio-Inspired Surface Modification to MEMS/Microfluidics System Integration"

_micromachines, 2019, doi:10.3390/mi10020101_

Round 1
Reviewer 1 Report
This review article goes into the details of creating passive and active self-cleaning systems. Although a broad view of surface fabrication and self-cleaning modalities were reviewed, the article does not go into enough detail regarding designing these systems. In addition, several other biomimetic (such as recent lubricant-infused coatings) or droplet transport mechanism seem to be missing from this review. I have the following major comments before recommending this paper for publication:
- Further explanations could be provided for passive self-cleaning surfaces. For example, what parameters are of importance? what angles (of the microstructures) are required for re-entrant and double re-entrant structures? What spacing or height between pillars is required? Where is any chemical modifications required? if so, is there a trend?
- You did mention that some required fluoro-polymer coatings (line 194), while some don’t (line 210), is there another chemical coating used or just the innate properties of the materials?
- Authors have failed to completely cover the recent advance4s in an important class of self cleaning coatings: lubricant-infused coatings. Authors should discuss the following papers in this review:
- Analytica chimica acta 1000, 248-255 (https://doi.org/10.1016/j.aca.2017.11.063)
- ACS Nano, 2018, 12 (11), pp 10890–10902 (DOI: 10.1021/acsnano.8b03938)
- AMI 5, 18, 2018 (https://doi.org/10.1002/admi.201800617)
- Scientific Reports 2017, (https://doi.org/10.1038/s41598-017-12149-1)
- Lines 54 -56: Discuss contact angle in relation of surface free energy, however, the equations are based on surface tensions. Is there an equation or relationship between surface tension and surface free energy? Please explain.
- Line 64: The word ‘improves’ is very subjective and does not explain how the surface wettability will change.
- Line 69: In equation 3, the authors demonstrate the case where a solid and air are present. Because a solid surface could be heterogenous in chemistry (or material), more terms could be present. This is not explained by the text making your explanation over-simplistic.
- Lines 81-82: Droplets on superhydrophobic surfaces do not slide. This is a misconception, as the true mechanism is more of a droplet rolling.
- Lines 92-93: It is unclear what the petal effect is, or why it has high CA hysteresis. Please explain in terms of advancing and receding angles.
Author Response
Dear Reviewers,
Thank you so much for reviewing our manuscript (manuscript ID: micromachines-404923). We sincerely appreciate your the time and effort. We have carefully reviewed the comments and have revised the manuscript accordingly. Please check the following document regarding our responses to the reviewers’ comments.
We hope the revised version is now suitable for publication and look forward to hearing from you in due course.
Best regards,
Di Sun, Karl Böhringer

Reviewer 2 Report
Response to manuscript micromachines-373040
Below you will find my impressions concerning the manuscript micromachines-373040 from Böhringer and Di Sun.
Impression
Form reviewer’s opinion the presented work is well written review about possibilities of self cleaing surfaces. But from reviewers opinion for a review article the number of references are to small and the theoretical details are very “limited” (without references the unfamiliar reader might be confused). From reviewers opinion the article can be accepted after minor revisions.
· You should mention a larger variety of literature and also indicate that “measuring” and “analyzing” CA (“who”, “conditions”) are controversial discussed: ADSA Neuman, Amirfazli, De Gennes, books of Mittal M. Schmitt and F. Heib, A more appropriate procedure to measure and analyse contact angles / drop shape behaviours, in Advances in Contact Angle, Wettability and Adhesion, Vol. 6, K.L. Mittal(Ed.), Wiley-Scrivener, Beverly, MA (2018). Or Edward Bormashenko, J. Drelich, Victor Strarov, Elliott, J.A.W.,Abraham Marmur, Schmitt (HPDSA) Adhesion of water droplet R. Tadmor (Lamar University). I have seen Butt, e.g. Young = thermiodynanamic equilibrium CA angle e.g. Schmitt, Heib, HPDSA. If youa re interested in energetic realtions I suggest Anatoly Rusanov or Possart, W.; Shanahan, M.E.R. Thermodynamics of adhesion. In Handbook of Adhesion Technology; da Silva, L. F. M.; Öchsner, A.; Adams, R. D., Eds.; Springer-Verlag, Heidelberg: 2011; Vol. 1, pp. 105-116.
From authors opinion controversiality is well explained in the contributions of Schmitt&Heib HPDSA and should be included in the theoretical part of this contribution to prevent any misunderstanding of the reader that the author (want to) covers the full scope of .
You should clearly state that you cannot cover the whole area and refer to newer references.
· Due to the fact that you mentioned the SLIP surfaces form reviewer opinion the “other way around” the liquid marble approaches of Edward Bormshenko might also be included or at least mentioned. But this is your decision and is not mandatory.
· 165 references?
· 195 references for fluoro-polymer coating? (spin coating of the substrate is not the only way.) – straight-forward vapor phase deposition can lead to “monolayers” Lu, X.L.; Munief, W…., R.; Ingebrandt, S. Front-End-of-Line Integration of Graphene Oxide for Graphene-Based Electrical Platforms. Adv Mater Technol-Us 2018, 3. - Munief, ---; Ingebrandt, S. Silane Deposition via Gas-Phase Evaporation and High-Resolution Surface Characterization of the Ultrathin Siloxane Coatings. Langmuir 2018, 34, 10217-10229. For information this group also performed CA investigations on model surfaces. If the fluoro-polymers are siloxanes this should be mentioned e.g. Analysis of silanes and of siloxanes are also investigated (influence of water, and catalyst). If other polymers are used these should also be mentioned including the references.
· Line 233 – for Photo-semiconductors like TiO2 not only the effect of oxygen should me mentioned. the reaction of the hole is also important. You might add something like “in combination with the Photo-Kolbe reaction [….], the photo induced oxidation/decarboxylation/fragmentation of organic acids which is well-known for photo semiconductors like TiO2 … ” Keyword Photo-Kolbe reaction: Kraeutler, B.; Jaeger, C.D.; Bard, A.J. Direct Observation of Radical Intermediates in Photo-Kolbe Reaction - Heterogeneous Photocatalytic Radical Formation by Electron-Spin Resonance. J. Am. Chem. Soc. 1978, 100, 4903-4905. Hoffmann, M.R.; Martin, S.T.; Choi, W.Y.; Bahnemann, D.W. Environmental Applications of Semiconductor Photocatalysis. Chem. Rev. 1995, 95, 69-96. Dolamic, I.; Burgi, T. Photocatalysis of dicarboxylic acids over TiO2: An in situ ATR-IR study. J. Catal. 2007, 248, 268-276. Schmitt, M.; Kuhn, S.; Wotocek, M.; Hempelmann, R. Photo-Curing of Off-set Printing Inks by Functionalized ZnO Nanoparticles. Zeit. Phys. Chem. 2011, 225, 297-311. Schmitt, M. ZnO nanoparticle induced photo-Kolbe reaction, fragment stabilization and effect on photopolymerization monitored by Raman-UV-Vis measurements. Macromol. Chem. Phys. 2012, 213, 1953-1962. Yang, D.; Ni, X.Y.; Chen, W.K.; Weng, Z. The observation of photo-Kolbe reaction as a novel pathway to initiate photocatalytic polymerization over oxide semiconductor nanoparticles. Journal of Photochemistry and Photobiology a-Chemistry 2008, 195, 323-329. Vautier, M.; Guillard, C.; Herrmann, J.M. Photocatalytic degradation of dyes in water: Case study of indigo and of indigo carmine. J. Catal. 2001, 201, 46-59. Schmitt, M. Synthesis and testing of ZnO nanoparticles for photo-initiation: Experimental observation of two different non-migration initiators for bulk polymerization. Nanoscale 2015, 7, 9532-9544.
Author Response

(The authors gave the same response as above.)

Reviewer 3 Report
In general, this paper is not well-organised. There are various issues, including missing references and poor quality of figures. English is not too big an issue, yet it needs to be moderately improved. While the application of liquid-repellent surfaces is of critical importance to everyday life, this review paper, unfortunately, does not manage to deliver any new messages compared with some previous publications. In fact, it is hard to follow the authors’ logic in choosing references, some of which cited by the authors within this paper do not necessarily best represent the best work in this field. What might help the authors to make their paper stand is to focus on the relation between liquid-repellent surfaces and MEMS. Considering the unmet need to integrate liquid-repellency into biomedical and microfluidic devices, it is highly recommended to turn this paper into an in-depth review thereupon to illustrate the progress and future potential of liquid-repellent (or self-cleaning) surfaces in relation to MEMS.
Abstract: The abstract is not clear to me. There are only two sentences in this Abstract, technically speaking, and the first one is too long to follow. The authors claim in the first sentence they “focus on self-cleaning surfaces”, without specifying which aspect(s). There are three main aspects interesting to researchers, (1) the natural models of self-cleaning surfaces, (2) the fabrication of artificial surfaces with controlled wettability, and the application of self-cleaning surfaces in engineering. Although the authors have mentioned some of the aspects in the second sentence, the question of “focus” has not been addressed.
L21: References are need for the definition of self-cleaning surfaces.
L26: Ref 1 is not even properly cited (otherwise it doesn’t exist). The title of the paper is “Natural and biomimetic artificial surfaces for superhydrophobicity, self-cleaning, low adhesion, and drag reduction”.
In addition, why is Ref 1 cited here alone? There are papers published in similar period of time, some even earlier, with even better clarifications and stronger significance.
L41-42: What are passive self-cleaning surfaces? References are needed, again, unless the authors are making original definitions.
L52: Define the subscripts in the equation.
L80: “Such a situation is categorized as superhydrophobicity.” High contact angle is one of the required conditions for superhydrophobicity. However, it doesn’t necessarily mean the surface will be superhydrophobic. In addition, the slide-off angle needs to be taken into account when superhydrophobicity is defined. Also, one needs to specify the type of liquid, which is, in this case, water, before referring to superhydrophobicity.
L164: Why is nature related to the low-surface tension liquids? They do not have high presence (apart from internal body liquids such as blood) in nature.
From L78 to L408: I do not see much point from these parts of the paper, apart from the fact the studies from the references are simply extracted and put together without additional intellectual input. The vast majority of references here are pre-2012 publications, which have been reviewed already many many times.
L339: The figure is not sufficiently explained in the text above it. There are publications discussing the role of ratchets in propelling water droplets, unfortunately none cited. The figure itself looks too busy. One can not easily get what it delivers. The impression is that this figure and those alike are meant for experts working on wetting dynamics. Yet this paper is meant for non-experts, in my opinion.
L362: The figure quality is too poor to be related to the text.
L396: Fig 13 has two sections, (a) and (b). The interpretations are too general. How do the Gecko setae help to inspire the artificial product? Any control experiments?
L404: Labels are needed to indicate the items.
Author Response

(The authors gave the same response as above.)

Reviewer 4 Report
This ms, which is submitted as a review paper accounts for the field of self-cleaning surfaces in a sober but also rather superficial manner with only few references cited and accounted for. Hence, deeper and more comprehensive review papers with many more references on the topic can be found in the literature already, e.g. Bharat Bhushan form 2011 (Natural and biomimetic artificial surfaces for superhydrophobicity, self-cleaning, low adhesion, and drag reduction), or Wen et al., Nanoscale 2017 (Biomimetic polymeric superhydrophobic surfaces and nanostructures: from fabrication to applications). Authors must include more references for this ms to become a comprehensive up to date review article on the topic.
Specific points:
1) Several statements are claimed without reference: l30-36, l79-86 (the conditions for various wetting states in terms of CA seem to be arbitrary, no references are given, and no justification is given). Here, it would here e.g. be appropriate to cite e.g. Nishino et al. "The lowest surface free energy based on -CF3 alignment" Langmuir (1999) 15, 4321–4323, which sets the scene for what can be obtained with flat surfaces.
2) Section 2.1 on theory is very superficial, missing e.g the Miwa model (Miwa et al. Effects of the surface roughness on sliding angles of water droplets on superhydrophobic surfaces. Langmuir 2000, 16, 5754–5760) for partially wetted structures. See also recent paper by Okulova et al. Nanomaterials 2018, 8, 831 and older papers by David Quéré et al regarding exprimental justification for the theoretical formulas. In addition, a discussion on validity of Cassie and Wenzel theory would be appropriate and distinguish this review from many others . See e.g. Gao, L.C.; McCarthy, T.J. How wenzel and cassie were wrong. Langmuir 2007, 23, 3762–3765, and papers that cited that publication.
3) L145-... Regarding jumping droplets, credit should also be given to Jonathan Boreyko,
"Self-Propelled Dropwise Condensate on Superhydrophobic Surfaces" Boreyko and Chen Physical Review Letters (2009), 184501 and Evelyn Wang, see e.g. Jumping-Droplet-Enhanced Condensation on Scalable Superhydrophobic Nanostructured Surfaces, Nano Lett. (2013), 13, 179−187.
4) L329-... on "anisotropic ratchet conveyors". This seems somewhat awkward to include this very specialized technique by one of the authors, having about 15 citation since 2012, while most other cited references in this ms are well known and highly cited publications.
Author Response

(The authors gave the same response as above.)

Round 2
Reviewer 1 Report
Authors have addressed most of comments. I suggest to accept the manuscript.
Author Response
Dear Editor, dear Reviewers,
Thank you so much for reviewing our manuscript (manuscript ID: micromachines-404923). We sincerely appreciate your time and effort. We have carefully reviewed the comments and have revised the manuscript accordingly. Our responses to the reviewers’ comments are shown in blue in the attached pdf.
We hope the revised version is now suitable for publication and look forward to hearing from you in due course.
Best regards,
Di Sun & Karl Böhringer
Reviewer 3 Report
After comparing the old and new versions of this review article, one can notice that the authors have made a few changes to improve the quality. However, the improvement is not adequate to justify publication. For example, the Abstract, which should be a concisely persuasive condense of an article, is still not strong enough in this case. The authors consider Ref 3 (old Ref 1) to be “a highly cited paper that provides a broad review of the field in the previous decade and earlier” and refuse to recognise any other references that have even better clarifications and stronger significance in the field. Here, I must make the point clear (also lengthy), since the knowledge of references is key to meaningful research. I can as least name three references (also review papers), which are not even published earlier, but also have been cited more:
Roach, Paul, Neil J. Shirtcliffe, and Michael I. Newton. "Progress in superhydrophobic surface development." Soft matter 4.2 (2008): 224-240.
Zhang, Xi, et al. "Superhydrophobic surfaces: from structural control to functional application." Journal of Materials Chemistry 18.6 (2008): 621-633.
Quéré, David. "Non-sticking drops." Reports on Progress in Physics 68.11 (2005): 2495.
Each of those three papers has significant contribution to the field. Roach et al.’s paper focuses on fabrication techniques and surface morphology of superhydrophobic surfaces. Zhang et al.’s paper focuses on a broad picture of superhydrophobic surfaces, from fabrication to application. Quéré’s paper follows his expertise in capillary and wetting dynamics (developed with de Gennes) to take on the challenges in understanding the wetting phenomena on superhydrophobic surfaces. On the other hand, Bhushan’s expertise is in tribology and friction. While a couple of his papers regarding superhydrophobicity have also been reasonably cited, Bhushan’s contribution to wetting phenomena on superhydrophobic surfaces is not yet comparable to that from many other pioneer scientists, including Aizenberg (Harvad), Varanasi (MIT), Stone (Princeton), Butt & Vollmer (MPIP). If one keeps updating his/her library of references, then they would notice for sure that these groups have been making greater contributions to this field. However, my point is not to criticise Bhushan’s work, but to help the authors to recognise better work in this field.
In addition, after comparing this very present review paper by Sun and Böhringer to the above mentioned papers, including Bhushan’s, I find nothing significantly new, apart from a handful of newer references. I understand that the authors would like to “explore both surface designs and their combination for MEMS/microfluidic systems for self-cleaning applications”, but it is unfortunately evident that this objective has not been delivered. What the authors could do is to bring more insight into MEMS/microfluidics that can distinguish this review from many others.
Author Response

(The authors gave the same response as above.)

Reviewer 4 Report
The authors have revised the ms appropriately.
Author Response

(The authors gave the same response as above.)
